# Novel Viruses in Mosquitoes from Brazilian Pantanal

**DOI:** 10.3390/v11100957

**Published:** 2019-10-17

**Authors:** Laura Marina Siqueira Maia, Andressa Zelenski de Lara Pinto, Michellen Santos de Carvalho, Fernando Lucas de Melo, Bergmann Morais Ribeiro, Renata Dezengrini Slhessarenko

**Affiliations:** 1Programa de Pós-Graduação em Ciências da Sáude, Laboratório de Virologia, Universidade Federal de Mato Grosso (UFMT), 78060-900 Cuiabá, Mato Grosso, Brazil; lauramsmaia@gmail.com (L.M.S.M.); andressazelenski@gmail.com (A.Z.d.L.P.); michellen.scarvalho@gmail.com (M.S.d.C.); 2Departamento de Fitopatologia, Instituto de Ciências Biológicas, Universidade de Brasília, 70910-900 Brasília, Distrito Federal, Brazil; flucasmelo@gmail.com; 3Departamento de Biologia Celular, Instituto de Ciências Biológicas, Universidade de Brasília, 70910-900 Brasília, Distrito Federal, Brazil; bergmann.ribeiro@gmail.com

**Keywords:** culicinae, high throughput sequencing, RNA virus, DNA virus, virome, sialovirome, phylogeny

## Abstract

Viruses are ubiquitous and diverse microorganisms arising as a result of interactions within their vertebrate and invertebrate hosts. Here we report the presence of different viruses in the salivary glands of 1657 mosquitoes classified over 28 culicinae species from the North region of the Brazilian Pantanal wetland through metagenomics, viral isolation, and RT-PCR. In total, 12 viruses were found, eight putative novel viruses with relatively low similarity with pre-existing species of viruses within their families, named Pirizal iflavirus, Furrundu phlebovirus, Pixé phlebovirus, Guampa vesiculovirus, Chacororé flavivirus, Rasqueado orbivirus, Uru chuvirus, and Bororo circovirus. We also found the already described Lobeira dielmorhabdovirus, Sabethes flavivirus, Araticum partitivirus, and Murici totivirus. Therefore, these findings underscore the vast diversity of culicinae and novel viruses yet to be explored in Pantanal, the largest wetland on the planet.

## 1. Introduction

Pantanal is the largest natural tropical wetland on Earth, encompassing 151,487 km^2^ in Brazil and small portions in Eastern Bolivia and Northeast Paraguay. This region is a large floodplain in which the hydro-ecological dynamics are regulated primarily by the flood-pulse, caused by excessive precipitation during the summer rainy season (November−March) followed by the drier winter season (April−September) interleaved by a transitional ebb period [1].

Pantanal represents an extremely diverse biome with the confluence of Brazilian Savannah (Cerrado), Amazon, Atlantic forest, and Bolivian Chaco. Currently, more than 650 terrestrial vertebrates species (96 reptiles, 40 amphibians, 390 birds, and 130 mammals) [1] and more than 9000 invertebrate species have been described in this biome [2,3], including a wide variety of hematophagous arthropods known as vectors of human and animal pathogenic arboviruses [4].

Taken together, these factors may contribute to arboviruses maintenance in Pantanal, making it a priority region for arbovirus discovery and surveillance in Brazil. Previous studies have demonstrated arbovirus serology and isolation in animals and mosquitoes, and also the isolation of novel insect-specific viruses in South Pantanal [5,6,7,8,9].

Emerging viruses have risen as a result of interactions between populations of hosts and pathogens and can potentially threaten the entire biodiversity. RNA viruses are considered the most potentially mutating and diverse viruses identified to date. High throughput sequencing (HTS) studies have boosted the discovery of several novel insect-specific viruses (ISVs) and arboviruses, enhancing our ability to access mosquito microbiome and amplifying the universe of viruses with poorly known modes of transmission and pathogenesis [10].

Culicinae mosquitoes are important human and animal infectious disease vectors worldwide, representing a public health problem [11]. Molecular and metagenomic approaches are important tools for studying the diversity of microorganisms present in insects, such as ISVs [10,12,13]. Surprisingly, these studies showed a vast diversity of ISVs that co-evolved and incorporated part of their genetic material in their host’s genome [14,15].

ISVs replicate exclusively in invertebrate cells and may interfere with vector susceptibility to arboviruses. Therefore, ISVs represent a possible biological mosquito control strategy to reduce arbovirus impact on human, veterinary and plant pathogen transmission [11,16]. Furthermore, these studies involving mosquito virome contribute with broad identification of viral diversity and evolution in different ecosystems and hosts.

ISVs are usually ancient viruses phylogenetically related to arboviruses, belonging to families *Flaviviridae, Togaviridae, Peribunyaviridae, Phenuiviridae, Rhabdoviridae,* and *Reoviridae*, and to non-arboviral families as *Mesoniviridae, Tymoviridae, Birnaviridae, Totiviridae*, *Partitiviridae, Iflaviridae, Chuviridae, Circoviridae,* and the taxon Negevirus [11,17,18].

Previous studies from our group already described new ISVs from *Chuviridae, Rhabdoviridae, Partitiviridae,* and *Totiviridae* families infecting the salivary glands of mosquitoes [19,20]. A novel phlebotomus fever serogroup member from *Phlebovirus* genus, *Phenuiviridae* family, named Viola virus was identified in *Lutzomiya longipalpis* from the High Pantanal region [21]. Therefore, this study aimed to identify the sialovirome of culicinae mosquitoes captured in High Pantanal, Mato Grosso State, Central-Western Brazil.

## 2. Materials and Methods

### 2.1. Mosquito Sampling, Processing, Random PCR, and Sequencing

Mosquitoes were captured during two consecutive days in five plots of a Rapid Assessment Program and Long Term Ecological Research (RAPELD) grid in Pirizal, High Pantanal (16°14′06”S, 56°22′70”W). We used three Nasci aspirators (13:00–18:00 h) for 30 min and five CDC light traps (18:00–06:00 h) at 1.5 m high along a transect with 50 m intervals during tree climatic periods (Figure 1). The insect capture in preservation areas has been previously approved by the Brazilian Environmental and Natural Resource Institute (SISBIO/ICMBIO) under the number 43909-1.

Briefly, captured specimens were kept alive under controlled temperature (24 °C), humidity, and artificial feeding with a 20% sucrose solution. Female mosquitoes were identified alive after immobilization (4 °C by 4 min) using dichotomy keys [22]; their dissected salivary glands [23] were pooled together (*n* = 3 to 117) according to date, place of collection, species, and gender; then homogenized in 0.4 mL of RNAse free phosphate saline buffer (pH 7.2) and centrifuged (5000× g for 4 min at 4 °C). RNA was extracted from the supernatant (0.2 mL) with a High Pure Viral RNA Kit (Roche) without RNA carrier, quantified (quantifluor RNA system; Quantus fluorometer, Promega, Madison, WI, USA), reverse transcribed (GoScript, Promega, Madison, WI, USA), and amplified in quintuplicates with a viral random PCR after double-strand cDNA synthesis (Klenow DNA polymerase I, New Englands BioLabs, Ipswich, MA, EUA) as previously described [19,21,24]. PCR products were purified with 20% polyethylene glycol, quantified with a quantifluor one dsDNA system (Quantus Fluorometer, Promega, Madison, WI, USA) and sequenced after genomic library preparation with the Truseq DNA PCR-free library kit (Illumina, San Diego, CA, USA) using 2 × 100 paired-end reads in two lanes with 60 GB on a Hiseq 2500 platform (Illumina, San Diego, CA, USA).

### 2.2. Genome Assembly, Taxonomic Classification, and Phylogenetic Analysis

Raw reads were quality trimmed and de novo assembled using CLC Genomics Workbench (v. 6.3, QIAGEN Bioinformatics, Aarhus C, Denmark). Contigs were then compared to a viral protein RefSeq database using Blastx [25] implemented in Geneious R11 (Biomatters, Auckland, New Zealand) [26]. All sequences with hits matching the viral database were additionally subjected to a Blastx search against the nr database. To confirm the assembly results and further extend incomplete genomes, trimmed reads were mapped back to the viral contigs and reassembled, until genome completion or no further extension. Final viral sequences were obtained from the majority consensus mapping assembly and annotated using Geneious R11 [26].

Viral amino acid sequences alignment was made with their corresponding homologs using MAFFT (Computer Systems Research Group from University of California, Berkeley, CA, USA) [27]. Phylogenetic trees were inferred by the maximum likelihood method (ML) implemented in FastTree [28], under the generalized time-reversible (GTR) model of nucleotide evolution + CAT model of amino acid evolution. Node support was determined using the Shimodaira-Hasegawa (SH) approximate likelihood ratio test [29]. Branch length represents the expected number of substitutions per site in the phylogenetic tree.

All trees were edited and visualized in FigTree (v1.4.3, http://tree.bio.ed.ac.uk/software/figtree). Viral sequences obtained in this study were deposited in GenBank (NCBI) under the accession numbers: MK780200, MK780201, MK780202, MK780203, MK780204, MK780205, MK780206, MK780207, MK780208, MK780209, MK780210, MN186291, MN186292, MN186293, MN186294, MN186295, MN186296, MN225577, MN225578, MN225579, MN225580.

### 2.3. Viral Isolation and Viral-Specific RT-PCR Design

Supernatants of pools positive for viruses in the HTS were inoculated (1:10 in L-15 medium) in C6/36 cells monolayers cultivated in T12.5 flasks containing L-15 medium supplemented with 10% fetal bovine serum and antibiotics/antimycotics for 2 h at 28 °C. Culture medium was replaced, and cell monolayers were maintained for five days at 28 °C. Then, cell monolayers were harvested for RNA extraction with Trizol LS (Thermo Fisher Scientific, Wilthan, MA, USA) and supernatant collected and stored at –80 °C.

Oligoucleotide primers were designed for iflavirus-like, circovirus-like, and rhabdovirus-like regions using the Primer3 available at Geneious R11 (Appendix A).

RT-PCR protocols targeted two flanking regions of the iflavirus-like (Ifla1F Ifla1R; Ifla2F and Ifla2R) and the M-G (MGMF and MGMR primers) and G-L gene regions of a rhabdovirus-like (GLMF and GLMR). These primers were used to reverse transcribe (Superscript IV, Thermo Fisher Scientific, Wilthan, MA, USA) total RNA extracted from cell monolayers (passage; p1) and viral RNA from the macerated salivary gland pool supernatant positive for those viruses.

Iflavirus-like sequences were amplified from 6 µL of cDNA in standard PCR reactions with the following conditions: 94 °C for 2 min, 35 cycles (region 1) and 30 cycles (region 2) at 94 °C for 1 min, 57 °C for 1 min, and 72 °C for 1 min and a final extension of 72 °C for 5 min. cDNA from Rhabdovirus-like regions were amplified in PCR reactions and cycled for 94 °C for 2 min, 40 cycles of 94 °C for 1 min, 50 °C for 1 min and 72 °C for 1 min and a final extension.

A PCR was also designed to amplify the circovirus-like (CircoF and CircoR) from the macerated pool with the following conditions: 94 °C for 2 min, 30 cycles at 94 °C for 1 min, 57 °C for 1 min and 72 °C for 1 min and a final extension of 72 °C for 5 min.

In addition, we used *Phlebovirus* segment M (primers PhleboMF and PhleboMR; 600 bp) and segment S (primers PhleboSF1, PhleboSF2 and PhleboSR; 400 bp) RT-PCR protocols to amplify the RNA extracted from the supernatant of the macerated pool and from the p1 monolayer of phlebovirus-like positive pools. Briefly, RNA was reverse transcribed (Superscript IV; Thermo Fisher Scientific, Wilthan, MA, USA) to cDNA (6 µL) and amplified in PCR reactions with the following conditions: 94 °C for 10 min, 55 cycles at 94 °C for 30 s, 35 °C (segment M) or 55 °C (segment S) for 1 min, and 72 °C for 2 min and a final extension of 72 °C for 10 min [30].

## 3. Results

We collected 1657 mosquitoes, 680 (41.0%) in the rainy period (mean temperature 31.8 °C and humidity 72.6%), 338 (20.0%) in the transitional (mean temperature 31.4 °C and humidity 86.1%) and 639 (39.0%) in the dry period (mean temperature 21.2 °C and humidity 64.4%). *Psorophora* (Ps.) *albigenu* was the most abundant species (1074; 65.0%) followed by *Mansonia wilsoni* (*n* = 142; 8.6%), *Aedes serratus* (*n* = 123; 7.4%), *Aedes scapularis* (*n* = 62; 3.7%), *Coquillettidia* (Coq.) *hermanoi* (*n* = 30; 1.8%), *Mansonia amazonensis* (*n* = 23; 1.4%), *Ps. ferox* (*n* = 18; 1.1%), *Wyeomyia* sp. (*n* = 16; 1.0%), *Aedes* sp. (*n* = 10; 0.8%), *Coq. venezuelensis* (*n* = 9; 0.5%), *Sabethes* (Sa.) *gymnothorax* (*n* = 3; 0.2%), and other 16 species (*n* = 147; 8.5%).

The raw reads from 10 sequenced pools/libraries (429,575,314 reads) were processed and assembled resulting in 239,485 contigs. Blastx comparisons against viral RefSeq database revealed twelve virus sequences (Table 1). Eight represent putative novel viruses named Pirizal iflavirus, Furrundu phlebovirus, Pixé phlebovirus, Guampa vesiculovirus, Chacororé flavivirus, Rasqueado orbivirus, Uru chuvirus, and Bororo circovirus and four already described mosquito viruses: Lobeira dielmorhabdovirus, Sabethes flavivirus, Araticum partitivirus, and Murici totivirus [19].

### 3.1. Iflaviridae

The genome (9367 nt) of a putative novel *Iflavirus* (Table 1) encodes one ORF (8,489 nt) flanked by two untranslated regions (5′ UTR 621 nt; 3′ UTR 311 nt) and a poly(A) tail. The ORF has three domains corresponding to picornavirus capsid protein (Rhv with 138 amino acids (aa) and Rhv-like with 145 aa within VP2 and VP3 regions, respectively), a capsid protein-like (CRPV with 210 aa, corresponding to VP1 protein) and an RNA-dependent RNA polymerase protein domain (RdRp with 571 aa).

This virus encodes a single polyprotein (2829 aa) with 33% aa identity with Wuhan fly 4 virus that possesses four structural proteins (capsid proteins: VP1, VP2, VP3, VP4) preceded by a leader region at the N-terminal region and the nonstructural protein: helicase, protease, and RdRp at the C-terminal region. Since isolates from the same species in this family share >90% of nt identity [31], this putative novel virus was named Pirizal iflavirus and was isolated in C6/36 cells at p1, confirmed by two RT-PCR protocols (Table 1). The phylogenetic tree (Figure 2) included Pirizal iflavirus in a clade with iflavirus Wuhan fly 4 virus and *Kinkell virus*, both ISVs [12,32].

### 3.2. Flaviviridae

Two sequences of *Flaviviridae* members were detected. The first (461 nt) codifies a region of the NS5 gene (69 aa) and shares 55% identity with the insect-specific *Menghai flavivirus* [33]. This virus presents less than 84% aa identity with other existing flaviviruses, criteria used to include novel flaviviruses [34], and was named Chacororé flavivirus (Table 1).

A second flavivirus sequence shares 94% identity with the insect-specific *Sabethes flavivirus* previously isolated from *Sabethes belisariori* in Brazil [35]. This sequence (1623 nt) codifies a region of the non-structural polyprotein encoding the NS2A (567 nt), NS3 (674 nt) and a region of the structural envelope protein (382 nt).

These viruses clustered distantly in the phylogeny, sharing a common ancestor with arboviruses from the flavivirus genus. Chacororé flavivirus was assigned in a cluster with *Menghai flavivirus*, whereas *Sabethes flavivirus* cluster is related to *Mercadeo* and *Calbertado*, all insect-specific flaviviruses (ISFVs) (Figure 2).

### 3.3. Phenuiviridae

Three partial genomic segments of a putative novel *Phlebovirus*, *Phenuiviridae* family were found in two pools of *Aedes scapularis* salivary glands (Table 1). The partial S segment (876 nt) codifies the nucleoprotein (291 aa), the M segment (144 nt) encodes a region of the structural polyprotein (47 aa), and the L segment (7553 nt) codifies the RNA dependent RNA polymerase (RdRp; 2,517 aa), with the characteristic phlebovirus RdRp domain (682 aa) and L protein N-terminus domain (68 aa). The RdRp, glycoprotein, and nucleoprotein regions of this virus presented 62%, 30%, and 40% of aa identity, respectively, with the most similar *Phlebovirus*, *Salarivirus Mos8CM0 virus* (accession numbers KX924627, KX924628, KX924629). This putative novel virus was named Furrundu phlebovirus.

Another L segment sequence (444 nt) belonging to this family shared 39% of aa identity with the most similar virus, *Severe Fever Thrombocytopenia Syndrome (SFTS) virus* (accession number: AGQ5425). This region codifies the RdRp (359 nt; 147 aa) and presents the RdRp (102 aa) domain. Therefore, this putative novel virus was named Pixé phlebovirus (Table 1).

Phleboviruses were confirmed by S and M segments RT-PCR protocols, amplified directly from the RNA obtained from the supernatant of the macerated pool and from passage 1 in cell culture (Table 1). RdRp phylogenetic tree suggests these viruses are classified into the severe fever with thrombocytopenia syndrome (SFTS) group, Uukuniemi serogroup, *Phlebovirus* genus, along with *Huaiyangshan, Malsoor* and *Salarivirus Mos8CMO viruses*. Based on L protein and M protein phylogeny, Pixé phlebovirus formed a monophyletic group with Furrundu phlebovirus and Salarivirus Mos8CMO inside the SFTS group. Nucleoprotein phylogeny also placed Furrundu phlebovirus in a cluster inside Uukuniemi serogroup (Figure 3).

### 3.4. Rhabdoviridae

A genome corresponding to the N (1168 nt), P (290 nt), and L (929 nt) genes of a putative novel rhabdovirus designated Guampa vesiculovirus (Table 1) shared 67% aa identity with *Benxi bat virus*, a *Vesiculovirus* isolated from *Rhinolophus ferrumequinum* in China [36]. This virus contains two domains: the RdRp (Mononeg_RNA_pol) with 138 aa and the nucleocapsid (Rhadbo_ncap) with 244 aa. Guampa vesiculovirus sequence presents four genes (3′-N-P-G-L-5′) with two conserved domains, lacking the region that corresponds to the M gene. Regions of this virus were amplified from the macerated pool and from C6/36 cell monolayers at passage 1 (p1) by RT-PCR (Table 1).

Another rhabdovirus genome (11275 nt) concatenated from three pools of *Psorophora albigenu* salivary glands (Table 1) had high nt identity to Lobeira virus (96%–100% identity), detected previously in mosquitoes captured in Chapada dos Guimarães National Park, Mato Grosso, Brazil [19]. The genome organization of Lobeira consists of five genes (3′-N-P-M-G-L-5′).

Phylogenetic tree based on the L protein placed Lobeira virus within *Dielmovirus* genus group I and Guampa vesiculovirus within the *Vesiculovirus* genus (Figure 4).

### 3.5. Chuviridae

A putative novel chuvirus glycoprotein gene region (864 nt; Table 1) sharing 69% of aa identity with Kaiowa glycoprotein was named Uru chuvirus. In the phylogenetic analysis, this virus was included in a clade with other mosquito-related members of the family identified in Brazil: Kaiowa, Guato [6], Cumbaru, and Croada viruses [19] inside the single new genus of this family, *Mivirus* (Figure 4).

### 3.6. Reoviridae

A VP1 segment (2854 nt; ORF 2796 nt) shares 38% aa identity with the Hubei reo-like 14 VP1 segment [12] and codify 931 aa of the RdRp. This sequence contains the RdRp domain (380 aa) of a putative novel *Orbivirus* named Rasqueado orbivirus (Table 1), since novel viruses included in this family share >30% of identity in the VP1 sequence [37]. VP1 phylogeny placed Rasqueado orbivirus closely to orbiviruses obtained from other arthropods, as Hubei reo-like 14 virus and *Anopheles hinesorum orbivirus* (Figure 5).

### 3.7. Circoviridae

The genome (768 nt) of a putative novel circovirus (Table 1) encodes two ORFs, the replication associated protein (Rep) (144 aa) and presents the viral rep domain (78 aa) and the capsid protein (84 aa), sharing 78% and 72% identity with Culex circovirus-like, isolated from *Culex* sp. in California [38]. This virus was amplified from the macerated pool by PCR and viral isolation was negative after three passages in cell culture (Table 1).

As seen for other *Krikovirus* members, this circovirus, named Bororo circovirus, presents a putative stem-loop structure serving as origin of replication (ori) at the 3′ end and is phylogenetically related to *Culex-circovirus like, Circoviridae TM-6c, Bat circovirus*, and *Mosquito circovirus* in the proposed *Krikovirus* genus [39] (Figure 6).

### 3.8. Partitiviridae and Totiviridae

Ariticum virus, a *Partitiviridae* member, was found in the salivary glands of *Sabethes gymnothorax* (RdRp; 1299 nt) and *Psorophora albigenu* (1439 nt) from Pantanal. These sequences shared 99% of aa identity with the virus originally described in the salivary glands of *Culex* sp. and *Mansonia wilsoni* mosquitoes from Chapada dos Guimarães National Park [19].

Another *Psorophora albigenu* pool presented a sequence (RdRp 907 nt; 99% identity) of the totivirus Murici virus, also previously identified in *Mansonia wilsoni* captured in Chapada dos Guimarães Park [19] (Table 1).

## 4. Discussion

Pantanal represents the world’s largest tropical natural floodplain, with a heterogenic ecological niche abundant in biodiversity supporting complex host biological networks, which in turn influences virus maintenance and evolution [2,4]. Nevertheless, sialovirome from potential vector species collected in this extremely dynamic landscape has been scarcely explored.

Here we describe 12 viruses predominantly detected in the dry climatic period and infecting mosquitoes from the High Pantanal region; eight represent previously undescribed viruses. In total, these viruses belong to nine viral families: *Iflaviridae, Phenuiviridae, Rhabdoviridae, Flaviviridae, Reoviridae, Chuviridae, Circoviridae, Partitiviridae,* and *Totiviridae*, representing one single stranded (ss)DNA virus and 11 RNA viruses: seven negative ssRNA (ssRNA-), 3 ssRNA+, one double stranded (ds)RNA. To our knowledge, these novel viruses are probably all ISVs.

In our study, ssRNA- viruses were surprisingly more frequently detected than other RNA viruses (Table 1). RNA viruses are recognized as a major cause of illness in their hosts when compared to DNA viruses. ssRNA+ viruses are more ancient viruses than dsRNA and ssRNA-viruses [40].

Recent studies have shown that ISVs are more ancient viruses when compared to arboviruses and have co-evolved with their mosquito hosts [14,15]. Interrelationships among culicinae and their ISV and arbovirus represent an area in extensive expansion since insects are major reservoirs of RNA viruses and may carry a vast number of unrecognized species. Due to the phylogenetic proximity, it is presumed that old and highly diversified strains of ISVs evolved and discerned over time with their hosts by the exchange of genetic material. It is very likely that many arboviruses were ISVs that have acquired the ability to infect vertebrates from gradual evolution [14,15].

Arboviruses are more frequently transmitted to mosquitoes during blood meals in infected individuals, trespassing gut barriers to establish persistent infection in acinar cells from the salivary glands. Vertical and venereal transmission are more associated with arbovirus maintenance during interepidemic periods. ISVs are maintained in mosquito populations essentially by venereal, transovo, and transovarian transmission to their partners and offspring, respectively, in natural and experimental conditions [41,42,43].

Arthropods are vectors of economically or clinically important diseases to a wide range of organisms, including animals, humans, plants, and may be considered an important source of accurate sampling of viruses circulating in a given region. In this study, *Ps. albigenu* and *Sa. Gymnothorax* contributed with the largest sialovirome. *Psorophora* and *Sabethes* species have been involved in Mayaro and yellow fever virus transmission cycles. ISVs previously described in these mosquitoes include Mayapan, Arboretum, and Sabethes flavivirus [35,41,44,45].

A putative novel *Iflaviridae* member found in *Psorophora albigenu* and named Pirizal iflavirus clustered with viruses found in flies. Iflaviruses are small non-enveloped ssRNA+ viruses encoding a single polyprotein. These agents are related to a wide range of insect hosts and many of them are pathogenic, producing developmental and behavioral abnormalities or death [46,47].

*Chuviridae* is a relatively recent classified family from order Jingchuvirales. Comprised by a monophyletic group of bi-segmented or non-segmented ssRNA- viruses, most of them present circular topology and have been reported in a wide range of arthropods, including flies, ticks [14], and different mosquito species from Pantanal, Central-Western Brazil [6,19].

*Reoviridae* are non-enveloped dsRNA viruses subdivided into two subfamilies, *Spinareovirinae* and *Sedoreovirinae*, this later includes *Orbivirus* genus, which present 10 linear dsRNA segments and are distinguished from other reoviruses for their transmission cycle, which involves arthropods [37]. Several members of this genus represent a problem for veterinary and public health, as bluetongue and Changuinola viruses, respectively [48,49]. Rasqueado orbivirus is closely related to several other ISVs from this genus. Although these ISVs present a common ancestor with arboviruses, phylogeny allocated these groups into distinct clades. We identified the VP1 and VP3 of this new orbivirus, VP1 is a highly conserved region among these viruses and represents the best marker for orbiviruses classification [37].

*Rhabdoviridae* consists of ssRNA- viruses infecting plants, animals, arthropods, and humans [50]. We identified two rhabdoviruses in this study: Lobeira virus, classified in the group I of Dimarhabdovirus supergroup, dipteran-mammal associated rhabdoviruses within Dielmovirus genus group I [19], and Guampa vesiculovirus, a novel *Vesiculovirus* genus member. These viruses present the typical rhabdovirus genome organization and differed more than 26% in their L protein sequence from the remaining members of the family.

Flaviviruses comprise several ssRNA+ arboviruses with worldwide distribution and wide host range. These viruses are allocated into four main phylogenetic groups: ISVs, viruses transmitted by mosquitoes, viruses transmitted by ticks, and no known vector viruses. Chacororé flavivirus is a putative novel virus that clusters with classical insect-specific flaviviruses (ISFVs) group of the genus *Flavivirus*, previously identified in *Aedes* mosquitoes [34,45,51]. Another virus belonging to the ISFV group identified in our study was *Sabethes flavivirus*, previously detected in *Sabethes belisariori* from Ribeirão Claro, Brazil, demonstrating this virus has a broader geographical distribution [36]. ISFVs have been demonstrated to alter vector competence of mosquitoes to arboviruses belonging to flavivirus genus, influencing the transmission of public-health important arboviruses [11,52].

Phleboviruses consist of enveloped negative or ambissense tripartite ssRNA viruses allocated into two major groups: phlebotomous fever segrogroup and Uukuniemi serogroup. Several ISVs have been recently reported and classified within this genus. Our results demonstrated that both viruses identified in this study are ISVs belonging to the Uukuniemi serogroup, SFTV group.

Circoviruses are circular non-enveloped ssDNA viruses that have been largely detected in vertebrates. Metagenomic studies changed this view revealing that viruses from this family also infect invertebrates [53]. Bororo circovirus clustered with ISVs from *Krikovirus* genus [39]. Lobeira, Araticum, and Murici viruses were previously discovered in *Aedes albopictus*, *Mansonia wilsoni* and *Culex* sp. salivary glands, respectively, in a previous study from our group in a geographically close region [19]. We found these viruses associated with other mosquito species in High Pantanal, showing their distribution within Culicinae from Mato Grosso State.

Our results revealed a number of undescribed viruses in mosquitoes from the High Pantanal microregion, providing complementary perspectives to virus diversity and evolution in Central-Western Brazil. These ecological and evolutionary perspectives represent advances in comprehending sialovirome of culicinae, although it remains unclear how evolutionary forces operate to promote interaction across different biological scales. Phylogenetic analysis of these newly identified viruses, including divergent members of their respective viral families, provided us initial insights into their evolution and taxonomic classification.

## Figures and Tables

**Figure 1 viruses-11-00957-f001:**
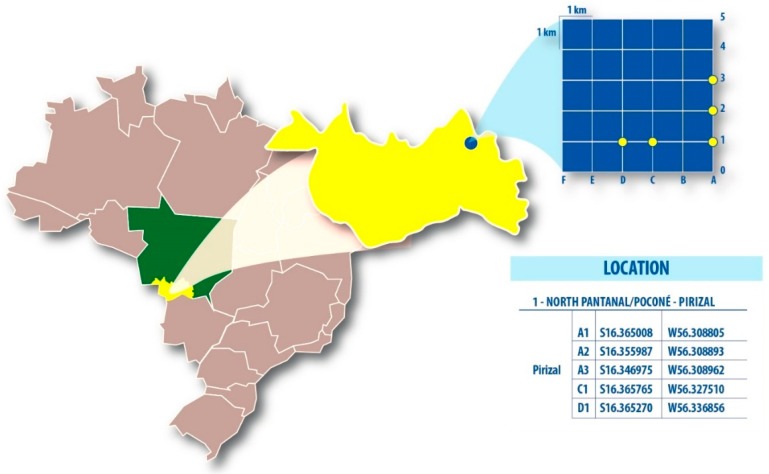
Rapid Assessment Program and Long Term Ecological Research system and the respective location of the five sampled grids in High Pantanal, Mato Grosso State, Central-Western Brazil.

**Figure 2 viruses-11-00957-f002:**
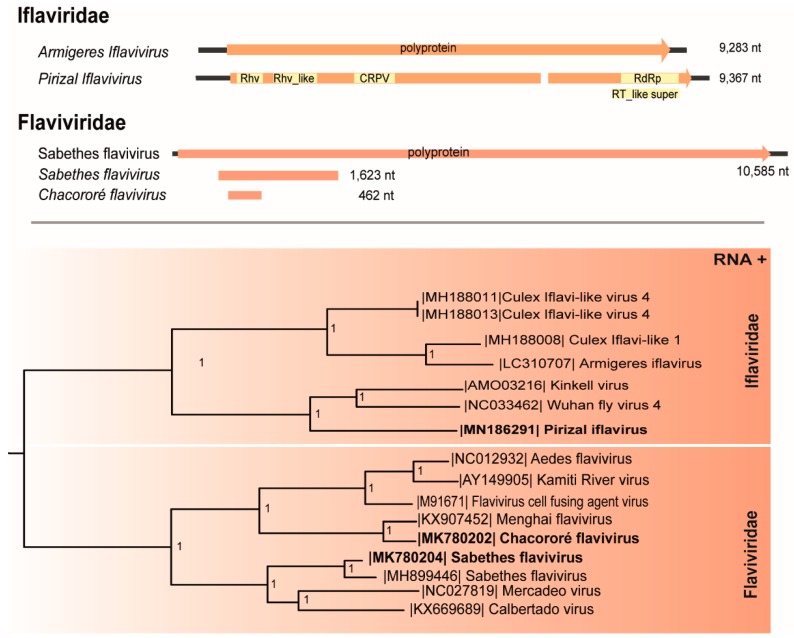
Genome map and maximum likelihood phylogenetic tree of Pirizal iflavirus, Chacororé flavivirus, and *Sabethes flavivirus* members of the *Iflaviridae* and *Flaviviridae* family, respectively. The length of each branch represents the expected number of amino acid substitutions per site.

**Figure 3 viruses-11-00957-f003:**
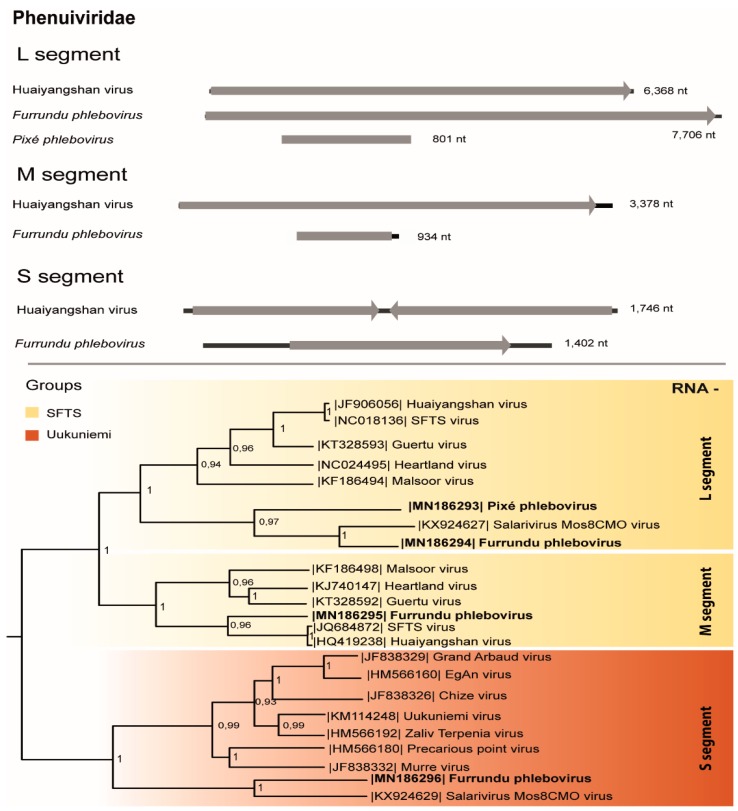
Genome map and phylogenetic tree of Furrundu phlebovirus and Pixé phlebovirus RNA dependent RNA polymerase (L), structural polyprotein (M), and nucleoprotein (S) by the maximum likelihood method and other *Phenuiviridae* members.

**Figure 4 viruses-11-00957-f004:**
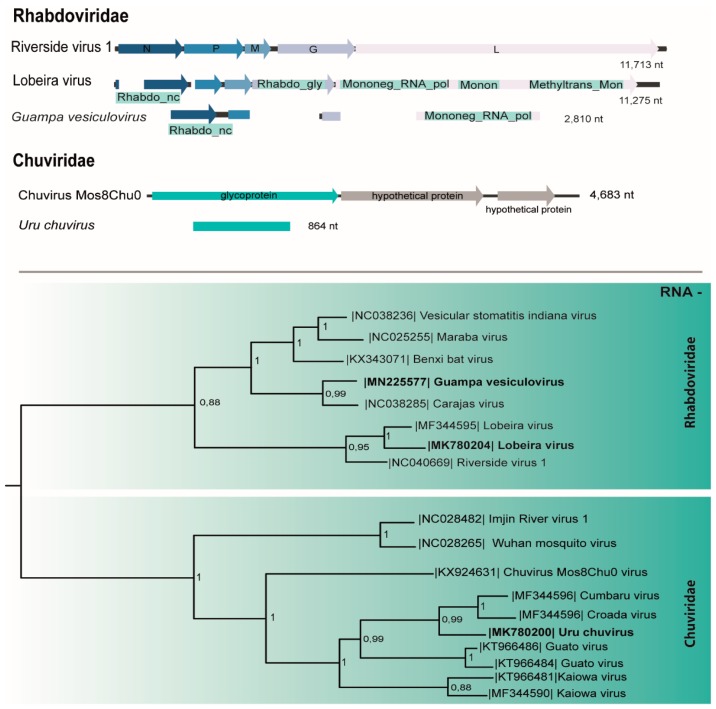
Genome map and the maximum likelihood phylogenetic tree of Guampa vesiculovirus, Lobeira virus, and Uru chuvirus members of *Rhabdoviridae* and *Chuviridae* families, respectively.

**Figure 5 viruses-11-00957-f005:**
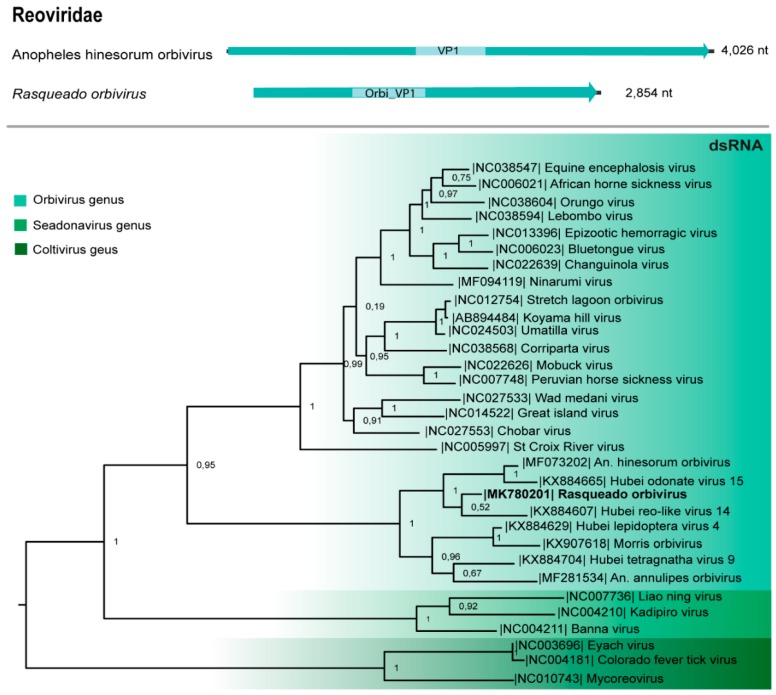
Genome map and maximum likelihood phylogenetic tree of Rasqueado orbivirus VP1 protein and other members of the *Reoviridae* family.

**Figure 6 viruses-11-00957-f006:**
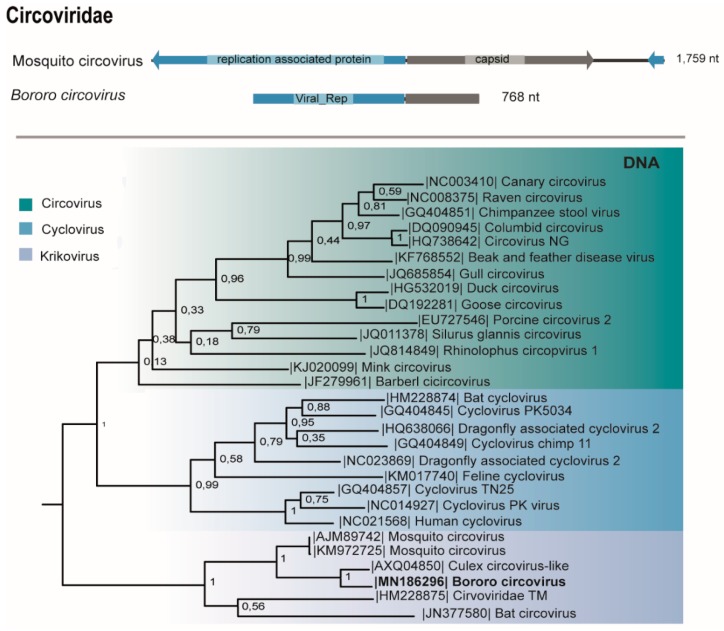
Genome map and the maximum likelihood phylogenetic tree of Bororo circovirus viral replication associated protein and other members of the *Circoviridae* family.

**Table 1 viruses-11-00957-t001:** Data obtained from culicinae pools captured in different climate periods at High Pantanal, Central-Western Brazil and subjected to high throughput sequencing.

Period	Plots	Pools	Species (n)	Viral Hit	Virus	Genome	Size (nt)	Identity (%)	E-value	Reads	Viral Isolation	Genbank
Rainy	A2; A3	M05, M06	*Psorophora albigenu* (100; 106)	*Rhabdoviridae (Dielmovirus)*	Lobeira virus	ssRNA-	11,275	100	0	24,452,482	Nt	MK780203
Transitional	A1	M14	*Psorophora albigenu* (110)	*Iflaviridae (Iflavirus)*	Pirizal iflavirus	ssRNA+	9367	33	3e-175	10,209,066	+p1, 300 pb	MN186291
	A1	M22	*Aedes (Ae) aegypti, Ae. Fluviatillis, Ae. crinifer* (5)	*Flaviviridae (Flavivirus)*	Chacororé flavivirus	ssRNA+	461	55	8e-37	13,768,232	nt	MK780202
Dry	D1	M25*	*Aedes scapularis* (7)	*Phenuiviridae (Phlebovirus)*	Furrundu phlebovirus	ssRNA-	L 7706	62	0	11,579,630	**500pb	MN186294
							M 934	30	2e-88			MN186295
							S 1402	40	7e-58			MN186296
	A3	M31	*Sabethes gymonothorax* (3)	*Flaviviridae (Flavivirus)*	Sabethes flavivirus	ssRNA-	1623	94	6e-95	17,431,974	Nt	MK780204
				*Chuviridae (Mivirus)*	Uru chuvirus	ssRNA-	864	69	3e-150			MK780200
				*Reoviridae (Orbivirus)*	Rasqueado orbivirus	dsRNA	2854	38	0			MK780201
				*Phenuiviridae (Phlebovirus)*	Pixé phlebovirus	ssRNA-	444	39	5e-21			MN186293
				*Partitiviridae (Unclassified)*	Araticum virus	dsRNA	1299	99	0			MK780207
	A2; A3	M33	*Coquillettidia albicosta, Coq. shannoni* (14)	*Rhabdoviridae (Vesiculovirus)*	Guampa vesiculovirus	ssRNA-	2387	67	2e-75	12,498,218	+p1, 300 pb	MN225577
	A1; A2	M35*	*Aedes scapularis* (11)	*Phenuiviridae (Phlebovirus)*	Furrundu phlebovirus	ssRNA-	L 7706	62	0	10,240,634	**500pb	MN186294
							M 934	30	2e-88			MN186295
							S 1402	40	7e-58			MN186296
	A1; D1	M37	*Psorophora. albigenu* (100)	*Totiviridae (Artivirus)*	Murici virus	dsRNA	907	99	0	13,477,440	Nt	MK780210
				*Rhabdoviridae (Dielmovirus)*	Lobeira virus	ssRNA	450	99	1e-96			MK780209
				*Partitiviridae (Unclassified)*	Araticum virus	dsRNA	1439	99	0			MK780208
	D1	M38	*Psorophora albigenu* (117)	*Circoviridae (Krikovirus)*	Bororo circovirus	ssDNA	768	78	6e-103	10,520,776	**760 pb	MN186292

* Same virus. N: number; nt: nucleotide; -: negative; +: positive; ss: single-stranded; ds: double-stranded; L: Large, M: medium, S: small segment. ** Detected directly from the macerated pool.

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
