# Peer review of "Novel Viruses in Mosquitoes from Brazilian Pantanal"

_viruses, 2019, doi:10.3390/v11100957_

Round 1

Reviewer 1 Report

In this manuscript from Siqueira Maia and colleagues, the authors collected salivary glands from mosquitoes from Pantanal, an important region for arbovirus discovery and surveillance due to its rich biodiversity, and conducted next generation sequencing on extracted RNA to characterise the virome of salivary glands. Their study detected eight novel viruses and four that were already characterised in previous studies. Their findings are a valuable addition to the collective understanding of the virosphere and should be disseminated. However, there are several issues that needs to be addressed before publication.

Major comments:

The motivation for focusing only on the salivary glands is not clear and confuses the aim of the study. Why are the salivary glands more important than any other tissue or even the whole body of the mosquito? If the goal is to detect ALL circulating viruses in Culicinae mosquitoes in Pantanal, whole mosquitoes would have allowed for a more complete and accurate insight. This and the study aim need to be explained unambiguously. The RNA extraction and library preparation methodology can be much improved with more detail: How many salivary glands were in a pool? If it is not the same for all pools, what are the minimum and maximum number of salivary glands? How were they homogenised i.e. in what volume of PBS? What primers were used in the reverse transcription following extraction with the High Pure Viral RNA Kit? The viral random PCR is an important step because it supposedly enriches viral sequences in the library. However, it is not clear how this step achieves this. The authors cite methods from a paper, which cited it from yet another paper. A brief summary of the protocol would be useful here. What is the read length of the Truseq DNA library kit used? 10 libraries were sequenced but no information was given about the composition or arrangement of these libraries. Were they randomised? Section 2.3, line 118-119 describe primers designed for only three virus families. Were RT-PCRs not done for the other viral families reported in the manuscript? If yes, what primers were used, i.e. for phlebovirus? Isolation and RT-PCR after sequencing is important to validate that detected viruses are live and actively replicating. Among the 12 viruses reported in this study, only two were successfully isolated and only five were detected via RT-PCR. This is something that should be addressed in the Discussion. How confident are the authors that the other viruses are actively circulating? Especially in the case of Chacororé flavivirus, of which only a contig of 462 nt was detected from sequencing data. What other possible explanations are there? The authors report RT-PCR results yet does not actually show any data. Given that the journal’s policy is to avoid using “data not shown”, I would suggest presenting these results as Supplementary Material at the minimum. The discovery of these new viruses is indeed significant and valuable for the field of virology but the authors overreach in their conclusion in stating that there are ‘complementary perspectives’ or ‘ecological and evolutionary perspectives’ from this study. There is no synthesis of new perspectives or ideas in the Discussion to support this claim. Perhaps a broader analysis, taking into account other studies, would be valuable i.e. what is the geographical spread of the four viruses already described elsewhere?

Minor comments:

Overall, the manuscript should be given a closer proof-reading. A few examples are: The first line of the Abstract is confusing as it suggests viruses are created by novel interactions between hosts and pathogens. Perhaps the author is referring to the diversity of viruses, instead of viruses themselves. Line 43: ‘potentially’ Line 54: ‘remain on’ should be replaced with ‘lie in’ Lines 165-168: the last two sentences of this paragraph should be swapped in position. Lines 215: ‘with 67%’ should be ‘shares 67%’ Virus genus names should be capitalised consistently. Not enough information is provided to describe the sequencing results (paragraph 2 of Section 3). How many contigs were found for each of the 12 detected viruses? Why was the sequencing divided up into 10 runs? What are the sizes of the largest continuous contigs? This would inform on whether the authors found intact viral genomes or fragments. In Figure 2, Pirizal Iflavirus is aligned with Armigeres Iflavirus but the latter is not introduced or explained. Why this virus? Was this the closest Blastx match for Pirizal Iflavirus? In Figure 3, should the background of the M segment section be red-coloured, based on the information in lines 202-209? It would be helpful to the reader if the authors added columns in Table 1 to indicate which viruses were successfully isolated from cell culture and which were detected via RT-PCR. Having horizontal lines to divide rows would also make it easier to read. Minor formatting issues in lines 242-244 and 253-256.

Author Response

REVIEWER 1

The motivation for focusing only on the salivary glands is not clear and confuses the aim of the study. Why are the salivary glands more important than any other tissue or even the whole body of the mosquito? If the goal is to detect ALL circulating viruses in Culicinae mosquitoes in Pantanal, whole mosquitoes would have allowed for a more complete and accurate insight. This and the study aim need to be explained unambiguously. -The mosquito salivary gland was used to avoid contaminations from legs, wings. We collected the salivary gland because this is the critical stage to stablish persistent viral infection with potential arboviruses. Arboviruses and viruses that infect the mosquito after ingestion must cross the intestinal barrier to reach the salivary gland, were they stablish a persistent infection in acinar cells. Therefore, we wanted to focus the study in those agents that were infecting the mosquito salivary gland and not other parts.

The RNA extraction and library preparation methodology can be much improved with more detail: How many salivary glands were in a pool? If it is not the same for all pools, what are the minimum and maximum number of salivary glands? Information inserted in the text. In the moment we developed the study, this is all our structure allowed us to develop.

How were they homogenized i.e. in what volume of PBS? Inserted in the text (400 ul). What primers were used in the reverse transcription following extraction with the High Pure Viral RNA Kit? The viral random PCR is an important step because it supposedly enriches viral sequences in the library. However, it is not clear how this step achieves this. The authors cite methods from a paper, which cited it from yet another paper. A brief summary of the protocol would be useful here. - Since we already detailed this methodology in previous articles (Pinto et al., 2017, Santos et al., 2018), we choosed not to repeat details and cite the reference.

What is the read length of the Truseq DNA library kit used? 10 libraries were sequenced but no information was given about the composition or arrangement of these libraries. Were they randomised? -  2 x 100 paired-end reads. Each library represents a pool of salivary glands according to date, place of collection, species and gender

Section 2.3, line 118-119 describe primers designed for only three virus families. Were RT-PCRs not done for the other viral families reported in the manuscript? If yes, what primers were used, i.e. for phlebovirus? - All the primers used in the experiment were included in this section. We did not performed RT_PCR for all of them, we designed primers only for rhabdovirus, iflavirus, circovirus and used a published protocol for phleboviruses, as indicated in the paragraph from line 134-140. Previous known viruses were not re-amplified.

 Isolation and RT-PCR after sequencing is important to validate that detected viruses are live and actively replicating. Among the 12 viruses reported in this study, only two were successfully isolated and only five were detected via RT-PCR. This is something that should be addressed in the Discussion. How confident are the authors that the other viruses are actively circulating? Especially in the case of Chacororé flavivirus, of which only a contig of 462 nt was detected from sequencing data. What other possible explanations are there? The authors report RT-PCR results yet does not actually show any data. Given that the journal’s policy is to avoid using “data not shown”, I would suggest presenting these results as Supplementary Material at the minimum. 

- During the experiment, we experience a severe fungal contamination in our cell culture. We had to change our biological safety cabinet, and dispose everything else to get rid of the contamination. Mean while, our TEMs were not successful and we spent all of the remaining volume we had from the macerated supernatant trying to isolate all viruses. We were successful only with the rhabdovirus and the iflavirus. For the flavivirus, we did not designed PCR primers because of the relatively small sequence we were able to detect. We therefore believe these viruses do exist and, although we do not have much data about them, is reasonable to include them in the publication as well and deposit the sequence we obtained, since in any other subsequential study we can find this virus again and obtain more data. Despite that, we have substantial data from the other viruses included in the manuscript. Some of them we were able to isolate and detect through RT-PCR. We did not take a good picture from the PCR products, that is the reason why we included as “data not shown”. Therefore, in accordance with the journal´s policy, we inserted this data and size of amplicon in the table as an additional column.

The discovery of these new viruses is indeed significant and valuable for the field of virology but the authors overreach in their conclusion in stating that there are ‘complementary perspectives’ or ‘ecological and evolutionary perspectives’ from this study. There is no synthesis of new perspectives or ideas in the Discussion to support this claim. Perhaps a broader analysis, taking into account other studies, would be valuable i.e. what is the geographical spread of the four viruses already described elsewhere? - The other viruses already described in this study were found in a very close region (i.e. 200 km of distance). We reformulated the conclusion removing “major”, “complement” is an appropriate word for our results and the described virus diversity in this rich but indeed poorly studied region. We have another ongoing experiment with more mosquitoes and vertebrates from the same location, this is a preliminary study, we have the objective to obtain more data implementing some improvements in methodology in this new subsequent project.

Overall, the manuscript should be given a closer proof-reading. A few examples are:

The first line of the Abstract is confusing as it suggests viruses are created by novel interactions between hosts and pathogens. Perhaps the author is referring to the diversity of viruses, instead of viruses themselves. – agreed, reformulated, novel removed.

Line 43: ‘potentially’ Line 54: ‘remain on’ should be replaced with ‘lie in’ - done

Lines 165-168: the last two sentences of this paragraph should be swapped in position.- done

Lines 215: ‘with 67%’ should be ‘shares 67%’ - ok

Virus genus names should be capitalised consistently. - done

Not enough information is provided to describe the sequencing results (paragraph 2 of Section 3). How many contigs were found for each of the 12 detected viruses? Why was the sequencing divided up into 10 runs? What are the sizes of the largest continuous contigs? This would inform on whether the authors found intact viral genomes or fragments. - Altogether it resulted in 239,485 contigs, the largest continuous contigs has 2.083 nt. Each run represents one pool of mosquito salivary gland, according to place, time of the year, species collected, gender. So we did not made a pool for sequencing, we sequenced each pool individually.

In Figure 2, Pirizal Iflavirus is aligned with Armigeres Iflavirus but the latter is not introduced or explained. Why this virus? Was this the closest Blastx match for Pirizal Iflavirus?  - Armigeres Iflavirus was one of the Blastx matches presenting complete genome sequence. Wuhan fly 4 virus and Kinkell virus are closely related to Pirizal iflavirus but both have incomplete genome sequences deposited at Viral RefSeq database.

In Figure 3, should the background of the M segment section be red-coloured, based on the information in lines 202-209? - corrected

 It would be helpful to the reader if the authors added columns inTable 1 to indicate which viruses were successfully isolated from cell culture and which were detected via RT-PCR. Having horizontal lines to divide rows would also make it easier to read. - Suggestion very pertinent and included in the text, in agreement also with reviewers suggestion to remore “data not shown” for these data in the main text. We do not have a good figure (PCR) that’s the reason why we did not included these results in the text.

Minor formatting issues in lines 242-244 and 253-256. – reviewed.

All the text was reviewed for fluency and grammar correction, as well as to let paragraphs more direct and linked with each other.

Reviewer 2 Report

The authors showed the finding of eight putative novel viruses with relatively low similarity with pre-existing species of viruses within their families. These findings underscore the vast diversity of culicinae and novel viruses yet to be explored in Pantanal, the largest wetland on the planet. However, the methods used are inadequate to achieve the objectives of the study. No full viral genome sequencing or virus isolation performed. 

Other comments

The article was written in small separated paragraphs especially the discussion with no connection or sound information flow.

Data presentation is very hard to understand

The figures legends are too short, the figures can't be understood

Author Response

REVIEWER 2

The methods used are inadequate to achieve the objectives of the study. No full viral genome sequencing or virus isolation performed. – actually we did and is presented in the results text. Iflavirus and Rhabdovirus are indeed almost complete genomes (we were unable to obtain short gaps inside de sequence).

The article was written in small separated paragraphs especially the discussion with no connection or sound information flow.

Data presentation is very hard to understand

The figures legends are too short, the figures can't be understood

-We tried to clarify the manuscript as most as we could, based also on the other reviewers observations.

Reviewer 3 Report

Authors performed NGS metagenomics for multiple samples from mosquitoes salivary glands homogenate to find and phylogenetically describe new viruses. Plenty of work have been done here. Unfortunately the part concerning phylogeny needs improvements and explanations. Without them I cannot recommend this interesting manuscript for publication in Viruses.

Text is well written but it need to be checked one more time with care. I found few minor mistakes.

45 – first appearance of an abbreviation “ISV” (not in lane 50). Please explain here.

61 – “taxon” should be “taxon”

73 – “mt” – Please explain the abbreviation

107 – “at” should be “in”

112 – “illumina sequencing”  - I do not prefer this kind of names for sequencing technology. If authors really want to stay with it please change to “Illumina sequencing” but it will be better just to use “NGS”.

123 – “invitrogen” change to “Invitrogen”

MAJOR:

Phylogeny:

I would like to know how exactly alignment for tree calculation has been prepared?  Why Authors used multiple models (GTR , GTR+I, JTT) and when? Please provide more detailed info why and when nucleotide or aa sequence have been used? Did Authors use only corresponding nt/aa in alignments or performed any curation? FastTree software is a very fast one but I am not sure if it is the best one. Authors should try to use MEGA software (if not for all analyses than just for few to confirm their phylogenetic trees). MEGA ( https://www.megasoftware.net/) is free and easy to work with but in Geneious software there are also other possibilities to build the tree.   

184 – and all phylogenetic trees. Please provide the legend and explanation for branch length.

Metagenomic analyses:

I admire how much work has been done by Authors in their pipeline, but in my opinion results could be better if true metagenomic software were used. Like KRAKEN (http://ccb.jhu.edu/software/kraken2/) or online tool http://kaiju.binf.ku.dk/server .

Author Response

REVIEWER #3

45 – first appearance of an abbreviation “ISV” (not in lane 50). Please explain here – corrected as suggested.

61 – “táxon” should be “taxon” – corrected.

73 – “mt” – Please explain the abbreviation – explained. Since it is not recommended to abbreviate words cited in the text less then 3 times, we corrected to “meters”.

107 – “at” should be “in” - done

112 – “illumina sequencing” - I do not prefer this kind of names for sequencing technology. If authors really want to stay with it please change to “Illumina sequencing” but it will be better just to use “NGS”. – we generally use HTS (high throughput sequencing) – corrected.

123 – “invitrogen” change to “Invitrogen”

MAJOR:

Phylogeny:

I would like to know how exactly alignment for tree calculation has been prepared?  Why Authors used multiple models (GTR, GTR+I, JTT) and when? – We use the Modeltest for each alignment. This information was corrected, since we used only the GTR model at the end.

Please provide more detailed info why and when nucleotide or aa sequence have been used?

Did authors use only corresponding nt/aa in alignments or performed any curation? Information provided in the results section. Phylogenetic analysis was always conducted with protein sequence.(Raw reads were quality trimmed and de novo assembled using CLC Genomics Workbench (v. 6.3). The resulting contigs were compared to a viral protein RefSeq database using Blastx [25] implemented in Geneious R11 [26]. All sequences with hits matching the viral database were then subjected to a Blastx search against the nr database).

FastTree software is a very fast one but I am not sure if it is the best one. Authors should try to use MEGA software (if not for all analyses than just for few to confirm their phylogenetic trees). MEGA ( https://www.megasoftware.net/) is free and easy to work with but in Geneious software there are also other possibilities to build the tree.

According to Price et al., 2010, the FastTree software has the same efficiency presented by other softwares, so we choose to use it, as our group is already familiar with this program and provided us the required licence.

184 – and all phylogenetic trees. Please provide the legend and explanation for branch length. – information inserted in the methodology section.

Metagenomic analyses:

I admire how much work has been done by Authors in their pipeline, but in my opinion results could be better if true metagenomic software were used. Like KRAKEN (http://ccb.jhu.edu/software/kraken2/) or online tool http://kaiju.binf.ku.dk/server. - These programs work well but often do not find all viruses. That's why we did it like this way. Thanks for the suggestion, we know these programs but we prefer to use our pipeline.

Round 2

Reviewer 2 Report

Manuscript is acceptable for publication. 

Reviewer 3 Report

No extra comment.